# Nanomechanical binding mechanism of ligands drives agonistic activity

Hannah Seferovic [1], Patricia Sticht[1], Lisa Hain[1], Rong Zhu [1], Sebastian Diethör[1], Christian Wechselberger[2,3], Florian Weber [4], David Bernhard[2,3], Birgit Plochberger [4], Yoo Jin Oh [1], Javier Chaparro-Riggers [5] & Peter Hinterdorfer [1] ✉

Monoclonal antibodies and ligands targeting CD40 exhibit a wide range of agonistic activities and antitumor responses. Studies have shown that the flexibility and affinity of antibodies play a crucial role in their immunostimulatory activity. However, a systematic comparison with the natural ligand is yet missing and a detailed investigation with respect to molecular rigidity, binding kinetics, and bond lifetime has not been undertaken to date. Here, we study the dynamic binding features of clinically relevant anti-hCD40 antibody subclasses, ChiLob 7/4, and the trimeric human CD40L to hCD40 at the single-molecule level. We visualize resembling of hCD40 receptors into dimers and higher-order oligomers that are dynamically captured and released by both ChiLob 7/4 and hCD40L with their multiple binding sites. Thereby, ChiLob 7/4 acts as a nanomechanical calliper and rotates its Fab arms in a highly dynamic fashion to screen for hCD40 binding, while hCD40L undergoes significantly less conformational changes. Despite its minor molecular flexibility, hCD40L performs association, dissociation, and re-association of hCD40 ten times faster when compared to ChiLob 7/4. We uncover a distinct binding mechanism that may explain the enhanced cluster formation potential and agonistic activity of the natural ligand and will inspire the design of novel ligand formats.

CD40[1,2], a member of the tumor necrosis factor receptor (TNFR) super family[3], is a co-stimulatory membrane receptor in antigen-presenting cells (APCs)[1,3,4] and tumor cells[2,4]. Stimulation of CD40 occurs through ligation of its physiological ligand, the stable trimeric form of CD40L, which is primarily expressed on surface activated CD4 T-lymphocytes[2,4,5]. The "pre-ligand assembly domain" (PLAD), located in the cysteine-rich domain 1 (CRD1) of the CD40 receptor, participates in CD40 dimerization and self-assembly[6,7], which facilitates multiple coupling to binding sites of CD40L for the formation of a final CD40 cluster[8,9]. Clustering of CD40 receptors in the APC membrane induced by the engagement of CD40 with CD40L is crucial in the activation and maturation of APCs and induces the release of cytokines that stimulate natural killer cells[2,3,10,11]. CD40/CD40L interaction also increases the surface expression of costimulatory and MHC molecules[10] and is necessary for the generation of cytotoxic T-lymphocytes[2,3,11].

Given their significant impact in immune regulation, CD40 and CD40L have become important targets in antibody-based cancer and autoimmune disorder therapies[2]. Anti-cancer immunity was enhanced from immunomodulatory anti-CD40 monoclonal antibodies (mAbs) by substituting CD4 T-lymphocytes and CD40L in the activation

[1]Institute of Biophysics, Johannes Kepler University, Linz, Austria. [2]Division of Pathophysiology, Institute of Physiology and Pathophysiology, Medical Faculty, Johannes Kepler University, Linz, Austria. [3]Clinical Research Institute for Cardiovascular and Metabolic Diseases, Medical Faculty, Johannes Kepler University, Linz, Austria. [4]Department of Medical Engineering, Upper Austria University of Applied Sciences, Linz, Austria. [5]BioMedicine Design, Pfizer Inc, San Diego, California, USA. ✉e-mail: peter.hinterdorfer@jku.at

process of APCs[5]. In addition, anti-CD40 mAbs showed a direct anti-tumor effect by initiating antibody-dependent cell-mediated cyto-toxicity (ADCC) or complement-dependent cytotoxicity (CDC)[2,5]. Several studies have demonstrated that the immunostimulatory activity of agonistic anti-CD40 antibodies depends on multiple factors. These factors include the location of the antibody's binding site on CD40[12] and the antibody's hinge flexibility, which inversely correlates with agonistic potency[13]. In addition, different IgG subclasses can induce varying CD40 receptor clustering[8,14] and antibody class switching has the potential to convert antagonistic into agonistic antibodies[14].

Human immunoglobulin Gs (hIgGs) are bivalent ligands and exist as four subclasses, IgG1, IgG2, IgG3, and IgG4, which vary in the composition and structure of their hinge region, affecting the antibody's flexibility. IgG3 is the most flexible, followed by IgG1, IgG4, and IgG2 as the least flexible[13,15]. IgG2 occurs in two isoforms, IgG2A and IgG2B[16,17], of which the compact structure of IgG2B is the most rigid among all subclasses[13,18]. IgG2B forms of anti-CD40 mAbs have demonstrated high agonistic activity even in the absence of FcγR crosslinking[8,12,14,18], although the Fc-FcγR coupling has been reported to be supportive[13,18]. Increased agonistic activity was also found for antibodies with lower affinities to CD40[19]. Differences in the disulphide-connectivity in the antibodies hinge region impacts on structure and function, such as on the superagonistic properties of immunostimulatory antibodies[18] and on factor VIII-mimetic activity[20]. However, a biophysical investigation with respect to flexibility, binding capability and dynamics has not been undertaken to date.

Here, we aim to decipher the mechanistic differences between CD40L and a range of agonistic anti-CD40 antibodies, which activity was found to depend on the IgG subclasses and the IgG2 isoforms[18]. We characterize a set of human IgG mAb subclasses (IgG1, IgG2, IgG4) and two isoforms of human IgG2, designated IgG-A, and −B[21], which differ by the disulphide-connectivity at the hinge region[16], and compare them with the trimeric form of the ligand CD40L. The key task is to paint a detailed dynamic and molecular mechanical picture upon ligation to our antigen candidates, isolated CD40[1,18] and CD40 stably expressed in CHO cells. We combine single-molecule force spectro-scopy (SMFS) on living cells with high-speed atomic force microscopy (HS-AFM) imaging to investigate, at single-molecule resolution, the interaction dynamics of the trimeric ligand and the bivalent antibodies to CD40. With SMFS, we deduce the interaction strength and dynamics during multi-bond formation. Using HS-AFM we film conformational dynamics of isolated CD40L and antibodies, as well as its complexes with CD40.

## Results

### Association and dissociation rates of hIgG subclasses binding to hCD40

To understand the binding mechanism and directly quantify the molecular interaction strength between hCD40 and monoclonal anti-hCD40 antibody ChiLob 7/4[22,23], we first performed SMFS experiments. Different antibody subclasses, including hIgG1, hIgG4, and the two isoforms of hIgG2 (hIgG2A and hIgG2B), all carrying the same binding paratope for hCD40, were tested. The antibodies were coupled to the AFM cantilever tip via a flexible polyethylene glycol (PEG) crosslinker[24] at their Fc region[25] (Fig. 1a). This linkage enables unconstrained binding of the antibody paratope to transmembrane hCD40 receptors

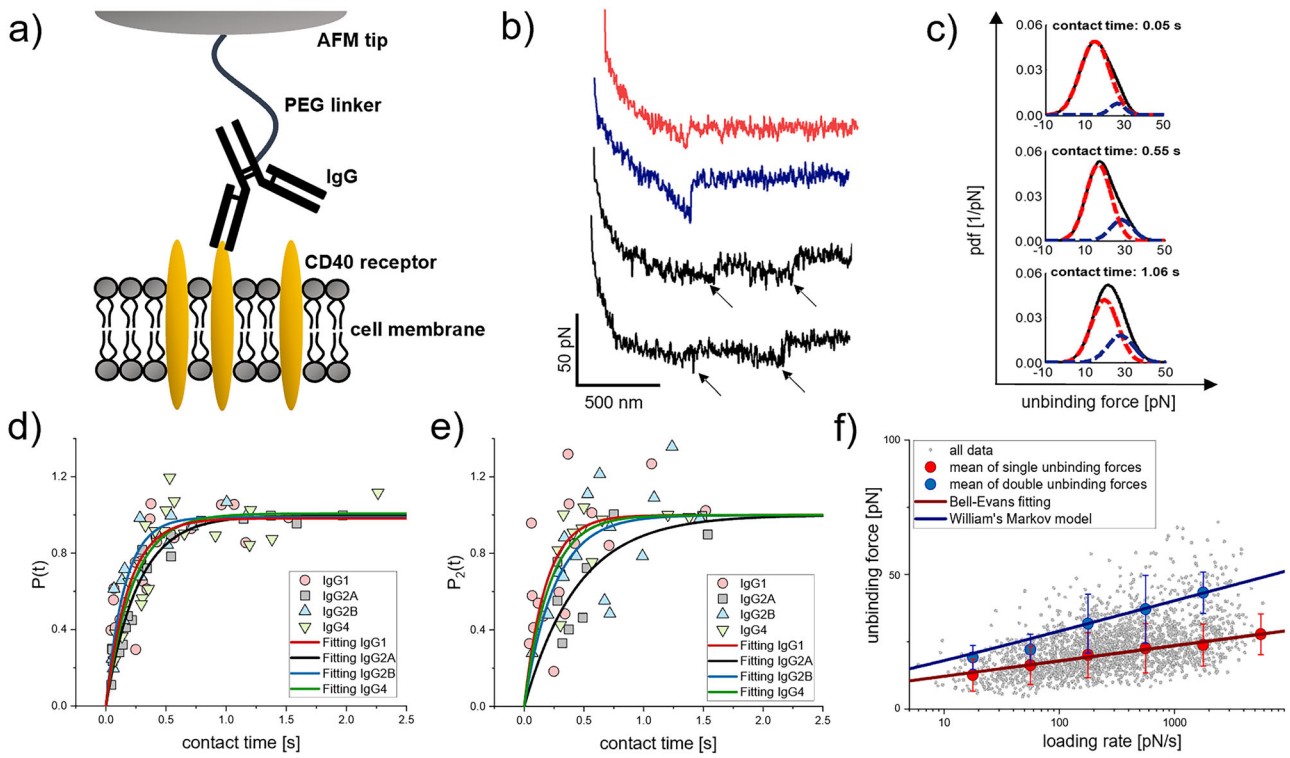

**Fig. 1 | Binding of hIgG subclasses to the transmembrane hCD40 receptor. a** Schematic illustration of SMFS experiments, in which the unbinding forces between hIgG, coupled to the AFM tip via a PEG crosslinker, and the transmembrane hCD40 receptor were measured. **b** Exemplary force-distance curves from the SMFS experiments with distinct rupture events due to single (red) hIgG:hCD40 bond dissociation, and simultaneous (blue) or sequential (black) dissociation of two hIgG:hCD40 bonds. **c** Experimental pdfs (black curves) of unbinding forces for different contact times. Pdfs peaks were fitted with multi-Gaussians, representing unbinding forces of one (red) and two (blue) hIgG:CD40 bonds. The frequency of two bonds increased with longer contact times. **d**, **e** Increase of probability of hIgG binding $P(t)$ and probability of second bond formation $P_2(t)$ over contact time, fitted using pseudo-first-order kinetics of a bimolecular reaction. **f** Unbinding force versus loading rate plot of hIgG2A. Data points (grey) were divided into loading rate segments. Mean and standard deviation of the first Gaussian (red dots) obtained from unbinding force pdfs of each segment were fitted using the Bell-Evans model (red fit). Higher forces were in good agreement with a Markov binding model (blue fit), predicting unbinding forces for parallel breakage of two identical bonds. Data presented are from three independent experiments.

**Table 1 | Summary of the kinetic rates $k_{off}$ and $k_{on}$, kinetic rates $k_{off,2}$ and $k_2$ for double bonds and affinities $K_D = k_{off}/k_{on}$ and $K_{D,2} = k_{off,2}/k_{on}$ for four different hIgG subclasses determined using SMFS**

| | $k_{off}$ [s$^{-1}$] | $k_{off,2}$ [s$^{-1}$] | $k_{on}$ [M$^{-1}$s$^{-1}$] | $k_2$ [s$^{-1}$] | $K_D$ [M] | $K_{D,2}$ [M] |
|---|---|---|---|---|---|---|
| **hIgG1** | $4.1 \times 10^{-2 \pm 0.26}$ | $2.7 \times 10^{-2 \pm 0.2}$ | $2.6 \times 10^{4 \pm 0.09}$ | $5.5 \times 10^{0 \pm 0.17}$ | $1.6 \times 10^{-6 \pm 0.26}$ | $1.0 \times 10^{-6 \pm 0.26}$ |
| **hIgG4** | $4.1 \times 10^{-2 \pm 0.15}$ | $2.7 \times 10^{-2 \pm 0.15}$ | $1.9 \times 10^{4 \pm 0.08}$ | $4.8 \times 10^{0 \pm 0.14}$ | $2.1 \times 10^{-6 \pm 0.15}$ | $1.4 \times 10^{-6 \pm 0.14}$ |
| **hIgG2A** | $2.9 \times 10^{-2 \pm 0.25}$ | $1.9 \times 10^{-2 \pm 0.25}$ | $1.7 \times 10^{4 \pm 0.04}$ | $2.2 \times 10^{0 \pm 0.13}$ | $1.8 \times 10^{-6 \pm 0.24}$ | $1.2 \times 10^{-6 \pm 0.25}$ |
| **hIgG2B** | $3.0 \times 10^{-2 \pm 0.16}$ | $2.0 \times 10^{-2 \pm 0.16}$ | $2.8 \times 10^{4 \pm 0.08}$ | $3.9 \times 10^{0 \pm 0.15}$ | $1.1 \times 10^{-6 \pm 0.16}$ | $7.2 \times 10^{-7 \pm 0.16}$ |

The errors of $k_{off}$, $k_{on}$ and $k_2$ are standard errors derived from least squares fitting. Other errors were calculated using error propagation.

expressed on the surface of CHO cells in a physiological setting and facilitates the detection of binding events mediated by the Fab arms of a single antibody[25]. We conducted consecutive force-distance measurement cycles[26] (Fig. 1b), during which a hIgG adorned to the cantilever tip was brought in contact with the hCD40 receptor on the cell surface to allow for bond formation. Thereafter, the AFM cantilever tip was retracted with defined speed such that the formed hIgG:hCD40 bond was loaded with an external mechanical force that increased over time. A typical force-distance cycle initially showed a nonlinear stretching behaviour resulting from the flexibility of the PEG cross-linker and the cell membrane. Finally, when a certain force was reached, the formed hIgG:hCD40 bond ruptured and the unbinding force was recorded (Fig. 1b).

The majority of the force-distance curves showed rupture events arising from the dissociation of single hIgG:hCD40 bonds (Fig. 1b, red curve). Double bonds, formed between the two Fab arms of the antibody and two hCD40 molecules, dissociated either simultaneously leading to an approximately two-fold unbinding force (Fig. 1b, blue curve), or sequentially (Fig. 1b, black curves). The probability for sequential dissociation correlated with the hIgG's flexibility arising from differences in the hinge region. Data analysis revealed $4.08 \pm 0.44$ % and $2.77 \pm 0.76$ % ($n = 2$ datasets) for the more flexible hIgG1 and hIgG4, respectively, and decreased rates for the less flexible hIgG2 isoforms, $2.42 \pm 2.30$ % (IgG2B) and $1.48 \pm 1.69$ % (IgG2A) ($n = 3$ datasets, $p > 0.11$). To verify that the recorded force signals indeed resulted from the dissociation of specific hIgG:hCD40 bonds, we performed control experiments on CHO cells lacking hCD40 expression (Supplementary Fig. 1a) and block experiments in the presence of soluble hCD40 in the bath solution of the measurement chamber (Supplementary Fig. 1b). In both cases rupture signals were largely absent.

To decipher the kinetics of bond formation we varied the contact time, in which the tip-adorned hIgG is in sufficient proximity to the cell membrane to allow for binding to hCD40. Unbinding forces for different contact times were collected and summed up in experimental probability density functions (pdfs) showing distinct force spectra (Fig. 1c, black). Fitting with sums of Gaussian resulted in bimodal force distributions and yielded an excellent reproduction of the data. All four hIgG subclasses showed a primary peak in the lower unbinding force region arising from monovalent hIgG:hCD40 bond dissociation (Fig. 1c and Supplementary Fig. 2, red Gaussian curve). The force maxima of the second peaks (Fig. 1c and Supplementary Fig. 2, blue Gaussian curve) confirmed that the dissociation resistance of double bonds is about 2-fold higher when compared to single bonds. Increasing the contact time enhanced the chance for double bond formation (Fig. 1c, blue Gaussian curves), which points out that hIgG forms two bonds sequentially, rather than simultaneously.

We then determined the kinetic association rate, $k_{on}$, for hIgG binding and the rate for formation of the second bond, $k_2$. Prolonging the tip-cell surface contact time led to an increase of both the overall binding probability, $P(t)$ (Fig. 1d, Supplementary Fig. 3a–d), and the probability for bivalent binding, $P_2(t)$ (Fig. 1e, Supplementary Fig. 3e–h). The plots were least-square fitted and $k_{on}$ and $k_2$ were computed by approximating with pseudo first-order kinetics according to $P(t) = A(1-exp(-(t-t_0)/\tau)$[27,28]. Similar values were obtained for all

four hIgG subclasses for the kinetic association rate, covering the range of $k_{on} = 1.7 \times 10^{4 \pm 0.04}$ M$^{-1}$s$^{-1}$ (hIgG2A) to $2.8 \times 10^{4 \pm 0.08}$ M$^{-1}$s$^{-1}$ (hIgG2B) (Table 1). The derived $k_{on}$-rates were in line with our surface plasmon resonance (SPR) experiments (Supplementary Fig. 4, Supplementary Table 1). With both SMFS and SPR, the rates for formation of the second hIgG:hCD40 bond, $k_2$, were higher for the more flexible hIgG1 and hIgG4 and lower for the hIgG2 isoforms. (Supplementary Fig. 3e–h, Table 1, Supplementary Table 1,). The greatest difference was found for hIgG1 and hIgG2A, $k_2 = 5.5 \times 10^{0 \pm 0.17}$ s$^{-1}$ (hIgG1) and $k_2 = 2.2 \times 10^{0 \pm 0.13}$ s$^{-1}$ (hIgG2A) derived from SMFS data, and $5.6 \times 10^{-4}$ RU$^{-1}$s$^{-1}$ (hIgG1) and $1.3 \times 10^{-4}$ RU$^{-1}$s$^{-1}$ (hIgG2A) determined using SPR (Supplementary Table 1).

To determine the kinetic dissociation rates, we then varied the retracting speeds in our SMFS experiments. Individual unbinding forces were plotted versus their loading rates (retraction speed times effective spring constant of cantilever, molecules, and cell) (Fig. 1f, Supplementary Fig. 5). From there the forces were divided into several loading rate segments (two segments per decade) and pdfs of unbinding forces of each segment were fitted with the sum of two Gaussians (Fig. 1c, Supplementary Fig. 2)[29]. Thereafter, the mean and standard deviation of the first Gaussian was plotted as a function of the loading rate (Fig. 1f, Supplementary Fig. 5, red data points) and fitted using the Bell-Evans model[30] (Fig. 1f, Supplementary Fig. 5, red fit), from which the kinetic dissociation rate, $k_{off}$, was derived (Table 1). $k_{off}$ values ranged from $k_{off} = 2.9 \times 10^{-2 \pm 0.25}$ s$^{-1}$ for hIgG2A, $k_{off} = 3.0 \times 10^{-2 \pm 0.16}$ s$^{-1}$ for hIgG2B to $k_{off} = 4.1 \times 10^{-2 \pm 0.1}$ s$^{-1}$ for hIgG4, and $k_{off} = 4.1 \times 10^{-2 \pm 0.26}$ s$^{-1}$ for hIgG1. Combining these rates with the kinetic association rates resulted in similar bond affinities, $K_D = k_{off}/k_{on}$, of $1.1–2.1 \mu M$, for all hIgGs (Table 1). Based on the fitting parameters of the Evans model, a Markov binding model[31] computed the theoretical forces for the dissociation of double bonds. The uncorrelated dissociation underlying this model implies that there is no mechanical coupling between the bonds. A good match with the mean and standard deviation of the second Gaussian in the higher force regime (Fig. 1f, Supplementary Fig. 5, blue fit, Table 1) revealed independent dissociation of the two Fab arms of hIgG from hCD40 molecules.

Together, our SMFS results show that the differences in the antibodies' hinge flexibility do not critically affect the formation and lifetime of single hIgG:hCD40 bonds (Table 1). Possible higher steric restrictions for hIgG2 isoforms are reflected in the lower number of sequential rupture events and in the slightly decreased rate for double bond formation as found by both SMFS and SPR (Supplementary Table 1). All four hIgG subclasses established stable bivalent bonds with two hCD40 molecules formed in a sequential binding mode.

## hCD40 self-assembles through the extracellular domain and establishes stable bivalent bonds with anti-hCD40 hIgG

To depict the dynamics of hIgG:hCD40 interactions in real-time, we filmed the molecules using HS-AFM under physiological conditions. Recordings of the extracellular hCD40 fractions on mica surfaces showed that they self-assembled into dimers and higher order oligomers (Fig. 2a, c). To estimate the number of oligomeric states, we determined the volume of individual hCD40 assemblies ($n = 111$) on the surface (Methods section 4.11). The volume data was then summed up

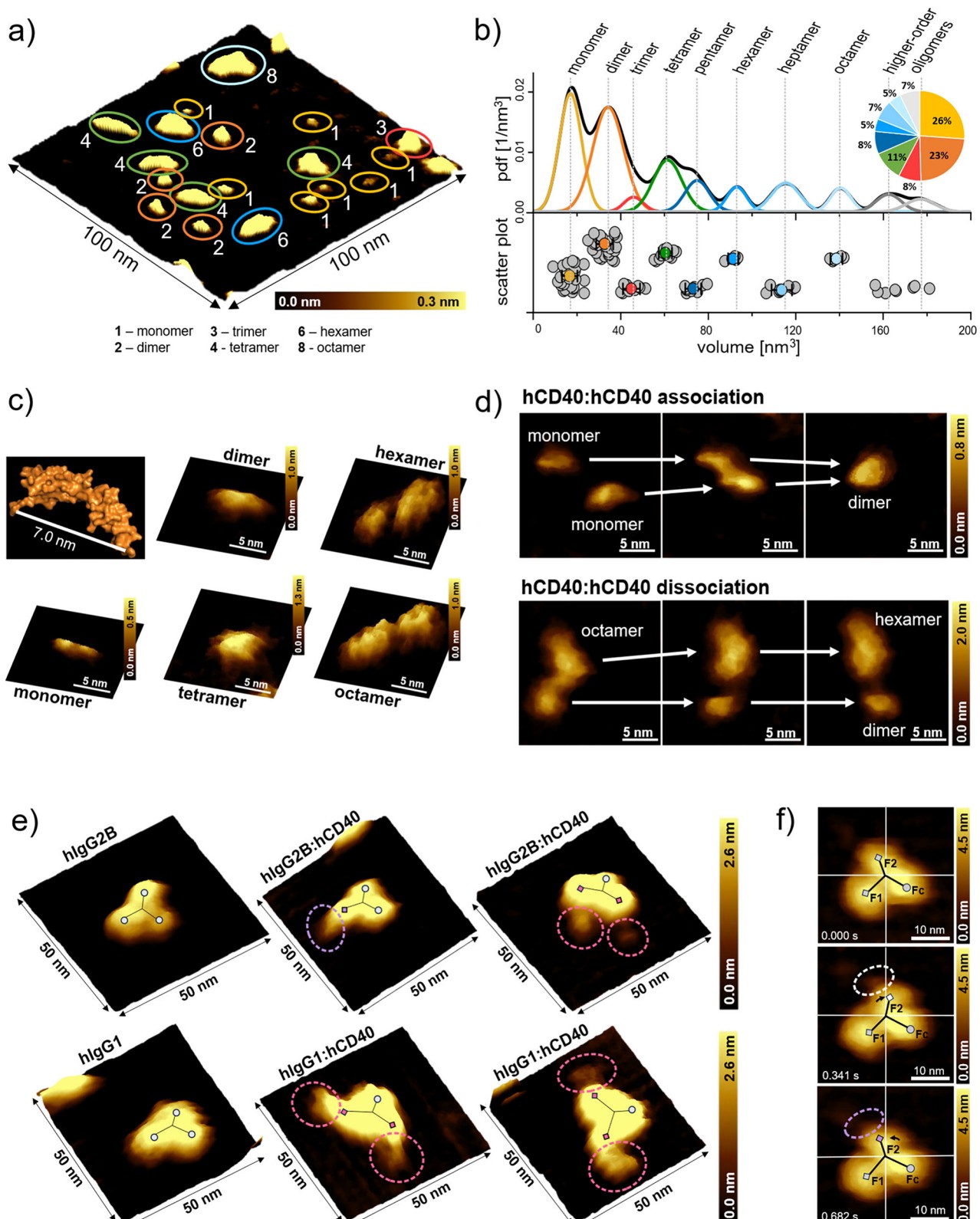

in an experimental probability function (pdf), as an equivalent of a continuous histogram representing the original data. The pdf was fitted with a multi-Gaussian function (Fig. 2b, pdf) and the individual Gaussians representing respective oligomeric states were differently coloured (Fig. 2b). Based on the maxima of all peaks (vertical lines in Fig. 2b) volume data points were collected from equidistant segments between two neighbouring peaks and presented in a scatter plot

(Fig. 2b), which allowed us to estimate the number of molecules in their oligomeric states: With 26% of monomers and 23% of dimers hCD40 appeared to be in an equilibrium between the monomeric and dimeric states. Other oligomeric states were less frequent with 8% of trimers, 11% of tetramers and lower proportions for assemblies of higher order (Fig. 2b, pie chart). The volume calculations were in good agreement with the known crystal structure of the extracellular CD40

**Fig. 2 | HS-AFM recordings displaying hCD40 oligomerization and its bond formation with hIgG. a** Image of hCD40 molecules of different oligomeric states recorded with HS-AFM. **b** Experimental probability density function (pdf) and its multi-Gaussian fit of the determined volumes of 111 different hCD40 assemblies showing two pronounced peaks, which can be assigned to monomers (yellow) and dimers (orange), accounting for nearly half of all hCD40 assemblies. The scatter plot depicts all volume data points (grey), which allows for estimating the number of oligomeric states. For each oligomeric state, the mean (colored) and standard deviation is displayed. **c** Crystal structure of the extracellular hCD40 monomer (PDB:3QD6) and 3D representations of HS-AFM images of hCD40 monomer, dimer and various oligomers. **d** Dynamics of the self-assembly of two hCD40 monomers into a dimer (first line) and disassembly of hCD40 octamer into a dimer and a hexamer (second line). The images are representative of hCD40 self-assembly and disassembly observed at multiple positions during the experiment. **e** 3D representations of HS-AFM images of hIgG2B and hIgG1 without (first column) and with hCD40 (second and third columns). hIgGs formed monovalent (purple) or bivalent (magenta) bonds with different hCD40 oligomers. hIgGs are marked by a y-shaped outline indicating the characteristic structure of the antibody's Fc region and its two Fab fragments. **f** HS-AFM images of hIgG showing the individual flexibility of the Fab arms (F1, F2) within a certain radius. Purple circles indicate monovalent hIgG:hCD40 bonds, white circles indicate an unbound state. hIgGs are marked by a y-shaped outline indicating the characteristic structure of the antibody's Fc region and its two Fab fragments. The images are representative of hIgG Fab arm flexibility observed across multiple antibodies during the experiment.

receptor (Fig. 2c). Self-assembly and dis-assembly of hCD40 receptors was highly dynamic, as evidenced from the association of two hCD40 monomers (Fig. 2d, first line, Supplementary Movie 1) and the dissociation of higher-order oligomers (Fig. 2d, second line, Supplementary Movie 2). The observed dimer formation on mica may be driven by the pre-ligand assembly domain (PLAD) of CD40[6–8]. However, we found higher order CD40 assemblies in various shapes (Fig. 1a), suggesting that their formation may result from a different mechanism than that initiated by PLAD.

For monitoring hIgG-induced hCD40 association, we chose two different antibodies, i.e. hIgG2B, the most rigid of our antibodies (Fig. 2e, first line) and hIgG1, the antibody with the highest Fab arm flexibility (Fig. 2e, second line). Due to a tip broadening effect, possibly caused by soluble hCD40 in the measurement chamber adhering to the cantilever tip, the dimensions of the hIgGs increased slightly (Fig. 2e, second and third columns) in comparison to the recordings of hIgG only (Fig. 2e, first column). Consistent with our SMFS results, our HS-AFM records clearly showed interaction of both antibodies with hCD40 in a highly dynamic fashion, either monovalent (Fig. 2e, first line, second column, Supplementary Movie 3) or bivalent with two individual hCD40 molecules (Fig. 2e, first line, third column and second line, second and third columns; Supplementary Movies 4–6). The formation of the hIgG:hCD40 bonds was not obscured by hCD40 self-assembly, as evidenced by the interaction with both, small hCD40 assemblies (Fig. 2e, second column; Supplementary Movies 3, 5) and larger hCD40 assemblies (Fig. 2e, third column; Supplementary Movies 4, 6), even though both the antibody binding site to hCD40 and the pre-ligand assembly domain are located in the cysteine-rich domain 1 of hCD40[6,12]. Association and dissociation with surface-immobilized hCD40 molecules, was often accompanied by movement of one Fab arm (Fig. 2g, Supplementary Movie 7). This highlights the antibody's hinge flexibility and its ability to rotate its arm within a certain radius, thereby enabling it to associate to molecules within this range. In line with our SMFS and SPR results, the HS-AFM recordings revealed a stable bivalent bond formation of hIgG to two individual hCD40 monomers or oligomers.

### Trimeric hCD40L establishes a maximum of two bonds with hCD40

We next compared the binding mechanism and kinetics of trimeric hCD40L with the bivalent hIgGs and performed SMFS experiments on CHO cells containing hCD40 (Fig. 3a). There, we coupled hCD40L to the PEG-ylated AFM cantilever via its $His_6$-tag. The recorded rupture events mainly carried the signature from the dissociation of single hCD40L:hCD40 bonds (Fig. 3b, green curve). Control experiments on CHO cells without hCD40 and block experiments using soluble hCD40 evidenced the specificity of our recordings (Supplementary Fig. 6). Double bonds either dissociated simultaneously (Fig. 3b, blue curve) or sequentially (frequency of $3.90 \pm 1.35$ % ($n = 3$ datasets)) (Fig. 3b, black curves). Although the number of double bonds increased with longer AFM tip – cell surface contacts (Fig. 3c), we never found third

peaks indicative of triple-bonds in the unbinding force pdfs, even after prolonged contact times (Fig. 3c, second and third pdf).

The kinetic association rate $k_{on} = 1.6 \times 10^{5 \pm 0.07}$ M$^{-1}$s$^{-1}$ (Fig. 3d, green data points and fit) and the rate for double bond formation $k_2 = 1.7 \times 10^{1 \pm 0.09}$ s$^{-1}$ (Fig. 3d, blue data points and fit) for hCD40L were markedly increased when compared to hIgGs (Table 2). Consistently, higher kinetic binding rates were also observed from our SPR experiments (Supplementary Table 1). Here we compared fits of the bivalent (Supplementary Fig. 7, solid line) and the trivalent (Supplementary Fig. 7, dashed line) model. Our results show that the bivalent model provided a much more accurate fitting of the data, suggesting that double bonds were the most prevalent binding conformation. In addition, hCD40L dissociation kinetics, $k_{off} = 3.5 \times 10^{-1 \pm 0.10}$ s$^{-1}$, (Fig. 3e) was ten times faster than for hIgGs (Table 2). With the kinetic on-rate as determined above, the single bond affinity of hCD40L, $K_D = 3.2 \times 10^{-6 \pm 0.11}$ M, was comparable to that of hIgG (Table 2). Conclusively, hCD40L showed similar affinity to hCD40, but, in stark contrast to hIgGs, markedly faster bond formation and dissociation.

To visualize dynamic features of the multivalent interactions of hCD40L with hCD40 in real-time, we first recorded movies of isolated hCD40L using HS-AFM. The globular-like feature of hCD40L resolved in AFM recordings was in good agreement with the crystal structure (Fig. 3f, first column, Supplementary Movie 8). We then added extracellular hCD40 to hCD40L and tracked the interactions with the hCD40L structures adsorbed onto mica over time. Due to a tip broadening effect, most likely caused by the soluble hCD40 in the measurement chamber adhering to the cantilever tip, the dimensions of hCD40L increased slightly (Fig. 3f, second and third column) in comparison to the recordings of hCD40L only (Fig. 3f, first column). Furthermore, in contrast to the crystal structure (Fig. 3f, first line), hCD40 assemblies were found to be lower in height than the hCD40L to which they were coupled (Fig. 3f, second and third column). This difference in height is attributed to the flexible chain-like structure of hCD40 (Fig. 2c), suggesting that hCD40 assemblies were not in an upright position, but rather in a lying position on the mica surface. hCD40 was observed to rapidly associate, disassociate and re-associate with hCD40L, revealing a highly dynamic binding behaviour (Supplementary Movies 9–12). Intriguingly, we found one (Fig. 3f, second column) and two (Fig. 3f, third column), but never three hCD40 molecules simultaneously binding to hCD40L (Supplementary Movies 9, 10). In line with our SMFS and SPR experiments, hCD40L establishes a maximum of two bonds with hCD40 at the same time, leaving the third binding site unoccupied. Similar as hIgG, hCD40L forms bonds with hCD40 of different oligomeric states (Fig. 3g, Supplementary Movies 11, 12), concluding that hCD40 self-assembly does not prevent hCD40L:hCD40 bond formation. Extending the incubation time to 45 min after pre-mixing soluble extracellular hCD40 with hCD40L, we also found large hCD40L:hCD40 clusters on the mica surface (Fig. 3h, Supplementary Movie 13), demonstrating that the extracellular domains of hCD40 and hCD40L independently drive cluster formation. In contrast, using the same experimental protocol, no such clusters were found for hIgGs.

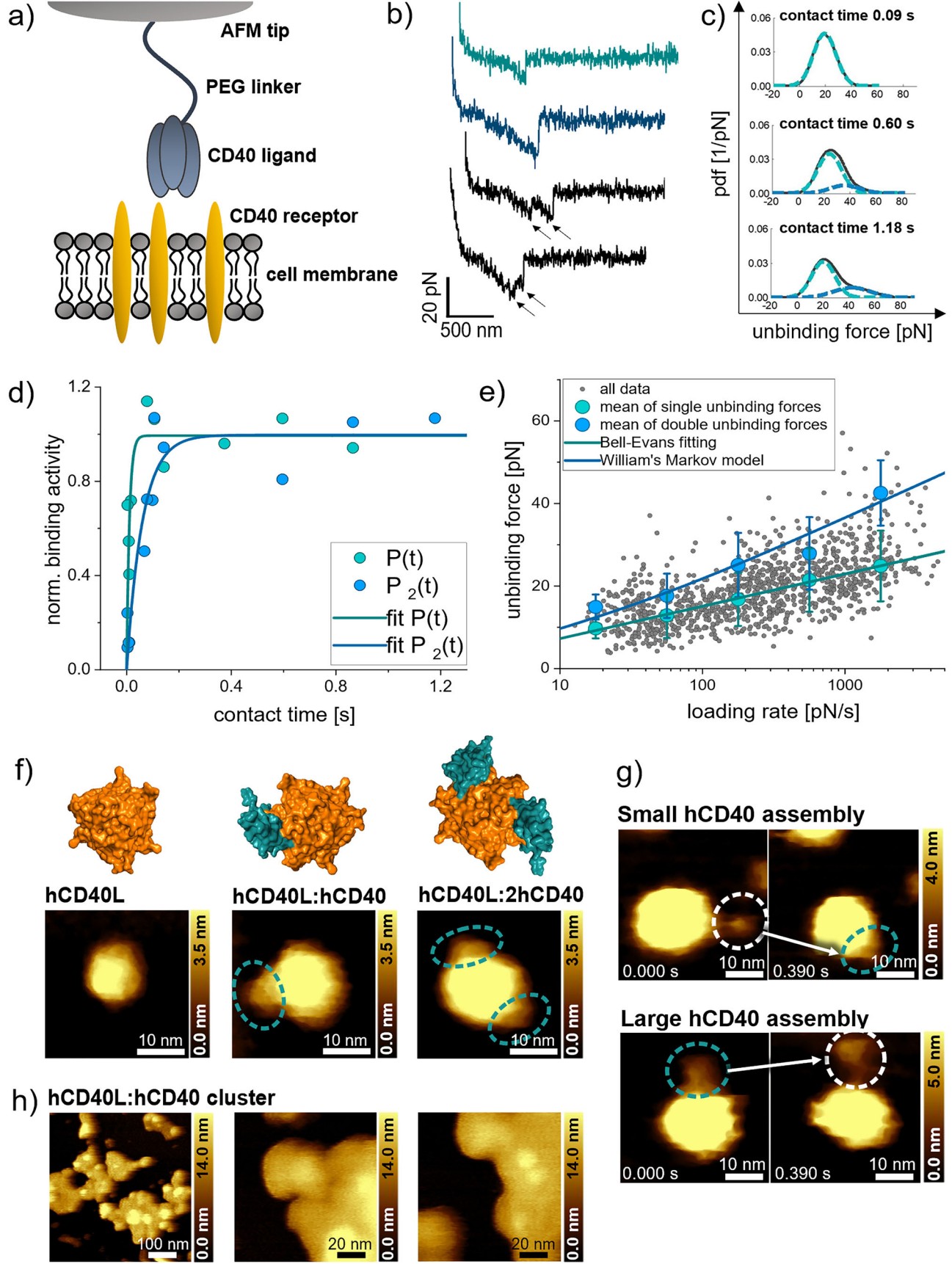

**Fig. 3 | Binding of trimeric hCD40L to hCD40 studied with SMFS and HS-AFM.**
**a** Schematic illustration of SMFS experiments of measuring the molecular interactions between hCD40L, coupled to the AFM tip, and the transmembrane hCD40 receptor. **b** Exemplary force-distance curves from the SMFS experiments with distinct rupture events arising single (green), simultaneous multivalent (blue) and sequential (black) hCD40L:hCD40 bond dissociation. **c** Pdfs (black curves) for different contact times: The peaks were fitted with Gaussian curves, summing up unbinding forces of monovalent (green) and bivalent (blue) hCD40L:hCD40 bonds. The number of double bonds increased with longer contact times. **d** Increase of binding probability over contact time for single (green) and double (blue) hCD40L:hCD40 bonds. Data points were approximated with a fit using pseudo-first-order kinetics to further derive the kinetic association rate constant and the association rate for two bonds. **e** Unbinding force versus loading rate plot: Data points (grey) were divided into loading rate segments. Pdfs of unbinding forces of each segment were created, and main peaks were fitted with Gaussian curves. The

Bell-Evans model (green fit) was applied to the mean and standard deviation of each Gaussian (green data points) to derive $k_{off}$. William's Markov binding model for double (blue fit) bonds was applied, predicting unbinding forces for two uncoupled parallel bonds. Data presented are from three independent experiments. **f** Crystal structures (PDB:3QD6) and HS-AFM images of hCD40L only (first column), with one (second column) and two (third column) hCD40 receptors bound. Of the three possible binding sites of hCD40L for hCD40, only a maximum of two were found to be occupied. The images are representative of hCD40L:hCD40 single and bivalent bond formation observed at several positions during the experiment. **g** HS-AFM images of association (first line) and dissociation (second line) of hCD40L with hCD40 assemblies of different sizes. Blue circles indicate hCD40 binding, white circles indicate an unbound state. Association and dissociation of hCD40L with hCD40 was observed at several positions during the experiment. **h** HS-AFM images of large hCD40L:hCD40 clusters formed on a mica surface. hCD40L:hCD40 clusters were observed at several positions during the experiment.

**Table 2 | Summary of the kinetic rates $k_{off}$ and $k_{on}$, kinetic rates $k_{off,2}$ and $k_2$ for double bonds, affinities $K_D = k_{off}/ k_{on}$ and $K_{D,2} = k_{off,2}/k_{on}$ for CD40L determined using SMFS**

| $k_{off}$ [s⁻¹] | $3.5 \times 10^{-1 \pm 0.10}$ | $k_{on}$ [M⁻¹s⁻¹] | $1.6 \times 10^{5 \pm 0.07}$ | $K_D$ [M] | $3.3 \times 10^{-6 \pm 0.11}$ |
|---|---|---|---|---|---|
| $k_{off,2}$ [s⁻¹] | $2.3 \times 10^{-1 \pm 0.10}$ | $k_2$ [s⁻¹] | $1.7 \times 10^{1 \pm 0.09}$ | $K_{D,2}$ [M] | $2.2 \times 10^{-6 \pm 0.10}$ |

The errors of $k_{off}$, $k_{on}$ and $k_2$ are standard errors derived from least squares fitting. Other errors were calculated using error propagation.

## hCD40 binds simultaneously with hIgG and hCD40L

The location of the binding site on hCD40 is critical in the agonistic activity of antibodies. Agonistic antibodies do not compete with hCD40L binding and restore activity, while antagonistic antibodies block hCD40L[12]. hIgGs (ChiLob 7/4) recognize an epitope on the cysteine-rich domain 1 on the N-terminus[12], whereas hCD40L binds to the opposite side of the extracellular domain. We tested the ability of hCD40L to form bonds with hCD40 in the presence of hIgG using a competitive SMFS binding assay. First, the binding probabilities of hIgG1 and hCD40L were monitored to ensure functionality (Fig. 4a). Then hIgG1 was injected into the bath solution at saturating conditions. We observed a clear reduction of the binding probability of the tip-adorned hIgG1 ($p = 0.052$) (Fig. 4b, grey data points). In contrast, the binding activity of the hCD40L tips remained unchanged ($p = 0.951$) (Fig. 4b, blue data points), although the hCD40 receptors on the cell membrane were fully occupied by hIgG1. Finally, addition of hCD40L lead to significant reduction of the binding probability ($p = 0.024$) and evidenced the specificity of the recordings from the hCD40L tips (Fig. 4c). Our results show that hIgG does not block hCD40L:hCD40 interaction. In fact, simultaneous binding of hIgG (ChiLob 7/4) and hCD40L to transmembrane hCD40 is essential for retaining the overall agonistic activity of both molecules at the same time.

## Discussion

Here, we studied the binding characteristics of different human IgG subclasses of the clinically relevant anti-human CD40 antibody, ChiLob 7/4, to CD40 on the single molecule level.

Crystallographic analysis of the ChiLob 7/4:hCD40 complex revealed that ChiLob 7/4 binds to an epitope near the N terminus of hCD40 at cysteine-rich domain 1 (CRD1), which is distal from the cell membrane and opposite from the hCD40L binding site of hCD40[12]. Our competitive binding assay using SMFS evidenced that ChiLob 7/4 association to hCD40 does not affect binding of the natural ligand hCD40L. Importantly, ChiLob 7/4 promotes hCD40 agonism and intensifies B cell activation and proliferation when applied with hCDL40[12], demonstrating that the antibody and ligand do not compete but retain agonistic activity together.

By utilizing HS-AFM, we identified soluble hCD40 mainly occurring in its monomeric and dimeric state. Formation of higher-order oligomers and dissociation occurred reversibly in a highly dynamically fashion. Association of hCD40 dimers was found to be routed through CRD1 via the pre-ligand assembly domain[6,7] located on the extracellular part of hCD40. Similarly, TNFR1, another member of the TNFR family, distributes in an equilibrium of monomeric and dimeric receptors, with PLAD initiating dimerization[9]. However, higher order CD40 assemblies were found in various shapes, suggesting that their formation may result from a distinct mechanism than that initiated by PLAD.

ChiLob 7/4 forms up to two stable bonds with hCD40 molecules independent of the oligomeric state of hCD40. Even though the binding site for ChiLob 7/4 and the pre-ligand assembly domain both lie within the CRD1 domain of hCD40, ChiLob 7/4:hCD40 association does not interfere with dimerization, nor does it disrupt pre-formed hCD40 oligomers. hCD40 pre-assemblies might facilitate the formation of clusters by bridging through neighbouring antibodies, as also found for TNFR1[9]. Notably, mAb binding to CRD1 leads to high agonistic activity, whereas mAbs binding to CRD2-4 exhibit weaker agonistic functions, possibly due to disruption of pre-formed hCD40 complexes that in turn prevent hCD40 clustering on the cell membrane and APC activation[12].

Among IgG subclasses, a higher agonistic activity of the IgG2B form of several anti-human mAbs including ChiLob 7/4 has been demonstrated[12,18]. To decipher how the nanomechanical and dynamic antibody properties drive hCD40 association and dissociation, we compared four hIgG subclasses using SMFS and derived the kinetic binding rates. No significant difference between the hIgG forms was found in monovalent bond formation and separation for all four hIgG subclasses (Fig. 5a, c, Supplementary Table 1, Supplementary Table 2). They established stable bivalent bonds with an average lifetime of $\tau = 43.73 \pm 8.26$ s, derived from the dissociation rate constant according to $\tau = 1/k_{off}$. Remarkably, the binding kinetics of the second Fab arm with another hCD40 molecule correlated with the antibodies' hinge flexibility and was faster for more flexible antibodies (Fig. 5b, Supplementary Table 1). However, as these increased kinetic binding rates were observed in both hIgG2A and hIgG2B, we conclude that they are not the primary driver of the enhanced agonistic activity that was specifically found for hIgG2B, being the most rigid isoform.

The interaction between the physiological ligand, the trimeric hCD40L, with hCD40 closely resembles the canonical pattern of the TNF-TNFR family interaction[1]. The binding site of hCD40 is located in a crevice formed between two hCD40L subunits and charge complementary plays a key role. Both, our HS-AFM and SMFS experiments revealed that at maximum only two bonds were formed with hCD40, leaving one binding site of hCD40L unoccupied. These finding agree with crystal structure analysis of the hCD40L:hCD40 complex, in which only two of the three structurally identical potential binding

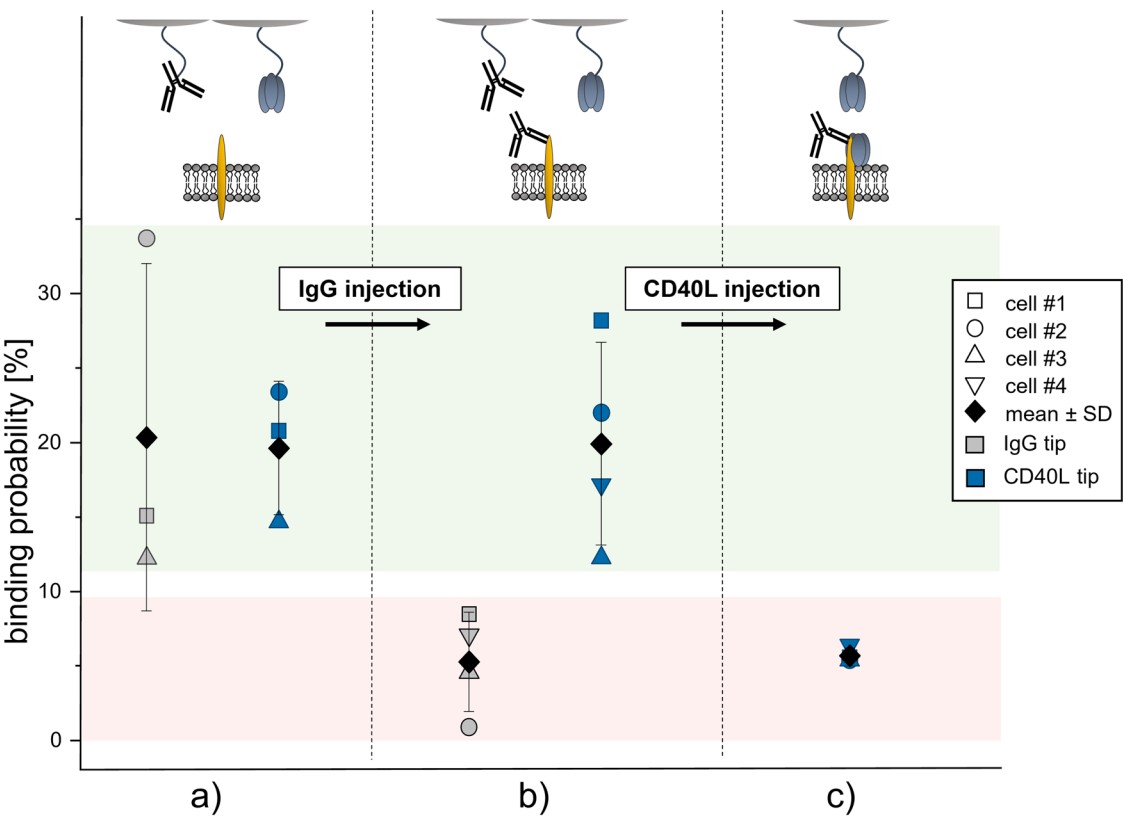

**Fig. 4 | Competitive binding assay for hIgG and hCD40L performed with SMFS.** **a** Binding probabilities of hIgG and hCD40L, coupled to the AFM tip, on cells expressing hCD40 receptors. **b** Addition of hIgG into the bath solution significantly reduces the binding probability of the hIgG-adorned tip, while the binding probability of hCD40L-adorned tip remains unchanged. **c** Additional injection of hCD40L also decreases binding of the hCD40L-adorned tip. Data were obtained from experiments on three to four different cells.

sites for hCD40 were engaged with the receptor[1]. While almost all TNFR family ligands in their trimeric conformation couple to three receptor molecules[32–36], there are other exceptions, such as LIGHT, which binds in a 2:3 binding ratio to its receptor LTβR under physiological conditions[37], and 4−1BBL with an atypical dimeric structure[38]. Our SMFS experiments revealed that hCD40L exhibits distinct kinetic characteristics when compared to ChiLob 7/4. Specifically, hCD40L shows both a larger dissociation rate and association rate constant, both of which differ by an order of magnitude from the rates derived for the hIgGs, ultimately resulting in comparable affinities (Fig. 5a–c, Supplementary Table 1, Supplementary Table 2). Despite the similar affinity and the faster dissociation of hCD40L from hCD40, hCD40L exhibits higher agonistic activity than ChiLob 7/4[8]. Counterintuitively, Yu et al.[19] recently found that anti-hCD40 mAbs with lower affinities showed increased agonistic activity and more potent antitumor activity than high-affinity antibodies[19]. The critical factor driving increased agonism was assigned to the higher dissociation rate constants. A similar correlation was depicted for anti-Fas antibodies, where the authors proposed[39] that IgGs first bind receptors bivalent, before one Fab arm dissociates for further recruiting receptors to finally establish a receptor cluster. Antibodies with slow dissociation rates might maintain their bivalent bonds, which impedes them from connecting to other receptors.

Using HS-AFM, we directly visualized the dynamics of ChiLob 7/4:hCD40 association and dissociation. Bivalent antibody binding to two hCD40 molecules was followed by dissociation of one Fab arm, before rapid re-association to another hCD40 molecule occurred, while the bond of the second Fab arm remained stable. The antibodies showed the ability to move their arms within about 10 nm of radius, allowing them to reach to hCD40 molecules further apart. This continuous dissociation and re-association mechanism could lead to more

and more receptors converging and forming a final cluster (Fig. 5d). We found that ChiLob 7/4 on average establish their second bond with a rate of $k_2 = 4.1 s^{-1}$, and a bond lifetime of $\tau = 29.2$ s. In contrast, hCD40L forms its second bond significantly faster, $k_2 = 17.0 s^{-1}$, and has a markedly lower bond lifetime, 2.9 s, which possibly allows it to rapidly recruit hCD40 receptors on the cell membrane (Fig. 5e, Supplementary Table 1, Supplementary Table 2). Additionally, the three binding sites of hCD40L that provide enhanced access to hCD40 receptors all around the ligand may further speed up receptor clustering (Fig. 5d, e). This idea is supported by our HS-AFM experiments, where we observed large hCD40L:hCD40 clusters, where in contrast no clusters were visible by hIgG. In conclusion, our study offers a thorough understanding of the binding dynamics of various hIgG subclasses and hCD40L with hCD40, along with the self-assembly of hCD40 receptors, which may support the development of effective ligands.

## Methods
### Antibodies and proteins
Stock solutions of hIgG1 (1 mg/mL), hIgG2A(C233S) (3.9 mg/mL), hIgG2B(C131S) (2.5 mg/mL), and hIgG4HS (hinge stabilized) (3.8 mg/mL) (Pfizer Inc., USA) were stored in PBS buffer (pH 7.4) containing 137 mM NaCl, 2.7 mM KCl, 8 mM $Na_2HPO_4$, and 2 mM $KH_2PO_4$ at −80 °C. C233S and C131S mutations in the hinge regions of hIgG2A and hIgG2B, respectively, prevent IgG2A/IgG2B class switching[40–42], whereas the hinge stabilization in hIgG4 prevents Fab-arm exchange[43,44]. Extracellular human CD40 (residues 20P-190D; 0.2 mg/mL) (Pfizer Inc., USA) with C-terminal $His_6$-tag was stored in PBS buffer (pH 7.4) containing 137 mM NaCl, 2.7 mM KCl, 8 mM $Na_2HPO_4$, and 2 mM $KH_2PO_4$ at −80 °C. Active trimeric human CD40 ligand[45] (hCD40L/hCD154; residues 116G-261L) with N-terminal $His_6$-tag (CDL-H5248) or N-terminal IgG1 Fc-tag (CDL-

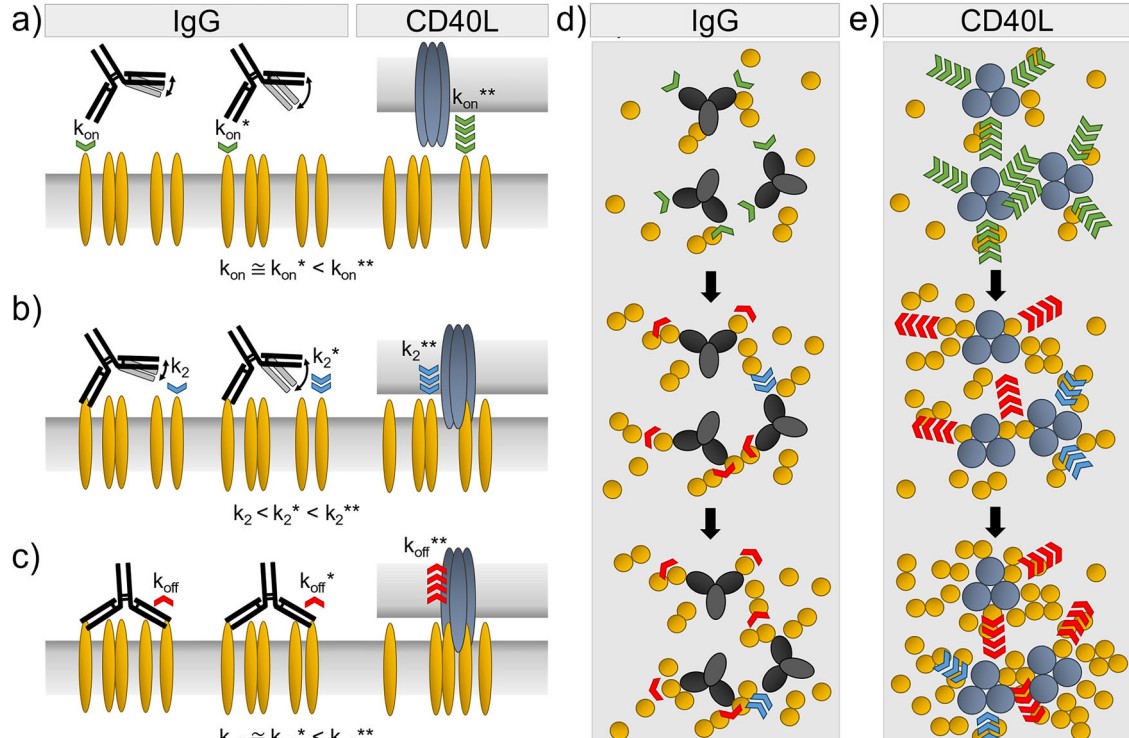

**Fig. 5 | Model for hIgG- and hCD40L-induced hCD40 clustering. a** Kinetic on-rates for single bond formation to hCD40 are similar for hIgG1 and hIgG4 ($k_{on}$) when compared to hIgG2A and hIgG2B ($k_{on}^*$), and ten times faster for hCD40L ($k_{on}^{**}$). **b** Kinetic on-rates for formation of the second bond to hCD40 are slightly slower for hIgG1 and hIgG4 ($k_2$), when compared to hIgG2A and hIgG2B ($k_2^*$), but markedly increased for hCD40L ($k_2^{**}$). **c** Kinetic off-rates for bond dissociation from hCD40 are similar for hIgG1 and hIgG4 ($k_{off}$), and for hIgG2A and hIgG2B ($k_{off}^*$). Kinetic off-rates are ten times faster for hCD40L ($k_{off}^{**}$). **d, e** Oligomerization of hCD40 driven by consecutive attachment, detachment, and fast re-attachment to binding sites of hIgG and hCD40L, respectively.

H5269) were purchased from AcroBiosystems, USA. hCD40L with N-terminal His$_6$-tag was stored in HEPES buffer (pH 7.4) containing 137 mM NaCl, 2.7 mM KCl, and 10 mM HEPES at −80 °C. Lyophilized hCD40L with N-terminal Fc-tag was reconstituted with 167 μL deionized water to a stock solution of 600 μg/ml, as described by the manufacturer, and stored at −80 °C.

## Human CD40 transfected CHO cells

CHO cells were transfected with Lipofectamine 2000 (Thermo Fisher Scientific Inc., #11668027) using a CD40 green fluorescent protein (GFP) fusion construct (Origene, #RG201977) to yield CHO-CD40-GFP cells. Cells were routinely grown on Petri dishes (Greiner Bio-One GmbH, #627 160) using DMEM (Gibco, #31966-021, high glucose, GlutaMAX™ supplement, pyruvate) containing 5% FBS (Gibco, #10500-064, qualified, heat-inactivated, E.U.-approved), 100 unit/mL penicillin and 100 μg/mL streptomycin (Gibco, #15140-122). Cells with high expression levels of CD40-GFP following G418-selection (Sigma Aldrich, #G8168) were enriched by using the GFP signal on a FACSAria™ sorter (BD Biosciences) and collected for further culture.

For AFM measurements, the density of the cells was about 10-30% coverage of the dish surface. The growth medium was exchanged to a physiological HEPES (4-(2-hydroxyethyl)−1-piperazineethanesulfonic acid) buffer containing 140 mM NaCl, 5 mM KCl, 1 mM MgCl$_2$, 1 mM CaCl$_2$, and 10 mM HEPES (pH 7.4 with NaOH).

Before force-distance curve measurements, the cells were examined by fluorescence microscope. A GFP filter set and an objective lens with a magnification of 10× or 20× were used to detect the expression level of CD40-GFP. Cells with very low expression were excluded from force measurements. Optimal cells for force measurements were those flat, smooth, and middle-sized cells with a homogeneous distribution of CD40-GFP. Usually, cells at the edge of a cluster were used, to

ensure that there was no other cell underneath the cantilever arms. Before transfer from the fluorescence microscope to the AFM, a mark point was made with a pen on the outside wall of the dish, so that the dish could be mounted in the AFM with the same orientation. With the optical microscope of the AFM, it was possible to find the selected cells.

## AFM tip functionalization with hIgG

5–10 μl of hIgG solution (Pfizer Inc.) (10 μl of 1 mg/ml hIgG1, 5 μl of 3.9 mg/ml hIgG2A, 5 μl of 2.5 mg/ml hIgG2B, and 5 μl of 3.8 mg/ml hIgG4) were diluted to 45 μl with 100 mM sodium acetate buffer (pH 5.5, adjusted with HCl) and dialyzed against sodium acetate buffer for 2 hours using a Slide-A-Lyzer MINI Dialysis Device with a MWCO of 10 kDa (Thermofisher, #69570). After dialysis, 5 μl of a freshly prepared sodium periodate solution (100 mM NaIO$_4$ in water) was added to the dialyzed antibody and was incubated for 1 hour at room temperature in the dark. Therefore, the Slide-A-Lyzer tube containing the antibody solution was placed above a glass beaker filled with water to prevent the drying of the protein sample. The treatment of antibodies with sodium periodate generates aldehyde groups on the two carbohydrate chains on the antibody's Fc arm and leads to a release of formaldehyde. To remove the formaldehyde, the IgG solution was dialyzed for 2 h against sodium acetate buffer and further dialyzed against fresh sodium acetate buffer overnight.

AFM cantilever tips (Bruker, MSCT) were washed 3 × 5 min in chloroform (VWR Chemicals, #22711.324) and dried with nitrogen. The cleaned cantilevers were amino-functionalized with APTES ((3-Aminopropyl)triethoxysilane) (Sigma-Aldrich, #440140) in the gas phase. Next, a Maleimide-PEG-NHS (Maleimide-polyethylene glycol-N-hydroxysuccinimide) linker was coupled to the cantilever tips by incubation for 2 hours in a solution containing 500 μl chloroform, 30 μl

triethylamine (Sigma-Aldrich, #90335), and 1 mg Maleimide-PEG-NHS linker. Next, the cantilevers were washed 3 × 10 min in chloroform and dried. To extend the maleimide group with a hydrazide function, cantilevers were immersed for 3 hours with a solution in which the components were mixed together in the following order: 50 μl water, 2 μl of 50 mM EDTA (pH 7.45), 5 μl of 500 mM HEPES (pH 7.4), 4 μl of 25 mM 3-[2-pyridyldithio]propionyl hydrazide (PDPH) in DMSO, 4 μl of 50 mM tris(carboxyethyl)phosphine (TCEP) hydrochloride, and 4 μl of 500 mM HEPES (pH 9.6). Next, the cantilevers were washed for 3 × 5 min in water and dried with nitrogen. The cantilevers were then covered with the hIgG solution (periodate treated and dialyzed before) and incubated for 2 hours. Finally, the cantilevers were washed 3 × 5 min with HEPES buffer containing 137 mM NaCl, 2.7 mM KCl, and 10 mM HEPES (pH 7.4, adjusted with NaOH) and were stored in HEPES buffer at 4 °C.

### AFM tip functionalization with hCD40L
AFM cantilever tips (Bruker, MSCT) were washed 3 × 5 min in chloroform (VWR Chemicals, #22711.324) and dried with nitrogen. The cleaned cantilevers were amino-functionalized with APTES (Sigma-Aldrich, #440140) in the gas phase. Next, a Maleimide-PEG-NHS linker was coupled to the cantilever tips by incubation for 2 h in a solution containing 500 μl chloroform, 30 μl triethylamine (Sigma-Aldrich, #90335), and 1 mg Maleimide-PEG-NHS linker. The cantilevers were then washed 3 × 10 minu in chloroform and dried. The cantilevers were incubated for 3 hours in a solution for which the components were mixed together in the following order: 50 μL of 2 mM thiol-tris-NTA (provided by R.Wieneke and R.Tampe., Institute of Biochemistry, Goethe-Universität Frankfurt), 1 μL of 100 mM EDTA (pH 7.5), 2.5 μL of 1 M HEPES (pH 7.5), 1 μl of 100 mM tris(carboxyethyl)phosphine (TCEP) hydrochloride, and 1.25 μL of 1 M HEPES (pH 9.6). Subsequently, the cantilevers were washed for 3 × 5 min in HEPES buffer containing 137 mM NaCl, 2.7 mM KCl, and 10 mM HEPES (pH 7.4, adjusted with NaOH). The cantilevers were then covered with a pre-mixed solution containing 100 μl of 1.5 μM His$_6$-tagged human CD40 ligand (hCD40L; AcroBiosystems, CDL-H5248) and 4 μl of 5 mM NiCl$_2$ and stored overnight at 4 °C. Before measurement, cantilevers were washed 3 × 5 min in HEPES buffer.

### Single-molecule force spectroscopy (SMFS)
SMFS experiments were performed on living CHO cells expressing hCD40 at room temperature using a commercial AFM (5500, Agilent Technologies, USA). Used cantilevers (Bruker, MSCT) had a nominal spring constant of 0.01 N/m, whereby exact spring constants were determined with the thermal noise method. The deflection sensitivity was determined from the slope of the force-distance curves recorded on a glass substrate to convert the output voltage signal into the cantilever deflection. Force-distance curves were acquired by recording between 500 and 1000 curves with a sweep duration of 0.5 s to 8 s and a vertical z-range of 3 μm, resulting in pulling speeds of 0.75 μm/s to 12 μm/s. Measurements with hIgG, coupled to the cantilever tip, were carried out in HEPES buffer containing 137 mM NaCl, 2.7 mM KCl, and 10 mM HEPES (pH 7.4, adjusted with NaOH), and force-distance curves were recorded with a hold time (time in which the cantilever tip remains on the surface) of 0.25 s. Measurements with hCD40L were carried out in the same HEPES buffer with an additional 50 μM NiCl$_2$. Additional measurements were performed with varying hold times at a pulling speed of 3 μm/s and 6 μm/s.

### SMFS data analysis
Recorded data sets were evaluated using a homebuilt Matlab-based software. For each unbinding event within a force-distance curve, the unbinding force, the effective spring constant, and the unbinding length were determined. In addition, for each data set of one pulling speed the binding probability, i.e., the number of unbinding events

over all recorded force-distance curves, was obtained. Next, unbinding forces of each complete data set of hIgG subclasses or hCD40L were plotted in a semi-logarithmic plot versus the loading rate, which is the product of the effective spring constant and the pulling speed. The data was divided with respect to the loading rate into 2 segments per decade with equidistant boundaries[29]. Segments containing less than 20 data points were excluded from further evaluation. Probability density functions (pdfs) of unbinding forces of each loading rate segment were constructed by summing up Gaussian areas computed for each unbinding force value, where the measurement error (standard deviation) of each Gaussian area was given by the noise amplitude of the corresponding force-distance curve. In the next step, the maximum peak of each pdf was fitted with a Gaussian function. Fitting of the means and standard deviations of each Gaussian curve in each loading rate segment was performed with a least-square fit using the Bell-Evans theory[30], whereby the width of the energy barrier $x_B$ and the kinetic dissociation rate $k_{off}$ were calculated. Forces of multiple parallel bond formation for different loading rates were obtained from William's Markov binding model[31]

$$r_f = k_{off} \frac{k_B T}{x_\beta} \left[ \sum_{l=1}^{N_B} \frac{1}{l^2} \exp\left(-\frac{F^* x_\beta}{l k_B T}\right) \right]^{-1} \tag{1}$$

with $r_f$ the loading rate, $F^*$ the unbinding force, $k_B$ the Boltzmann constant, $T$ the temperature, $N_B$ the number of bonds and $k_{off}$ and $x_B$ parameters for single bond formation obtained before with the Bell-Evans model. For the calculation of dissociation rates for simultaneous parallel bond rupture of two hIgG:hCD40 and two hCD40L:hCD40 bonds the William's Markov binding model was used with

$$k_{off, N_B} = k_{off} \left[ \sum_{l=1}^{N_B} \frac{1}{l} \right]^{-1} \tag{2}$$

Kinetic association rates were calculated with $k_{on} = (\tau c_{eff})^{-1}$ with the interaction time $\tau$ and the effective concentration $c_{eff}$, which was determined with $c_{eff} = N/(N_A V)$, where $N$ is the number of receptor molecules on the AFM tip, $N_A$ is the Avogadro constant and $V$ the volume of a hemisphere with a radius equal to the length of the PEG crosslinker and the length of the coupled molecule. The interaction time $\tau$ was determined by approximating the increase of binding probability $P$ over the contact time $t_c$ with $P = A(1-exp(-(t_c-t_O)/\tau))$, with $A$ the maximum binding probability and $t_O$ the lag time[27]. For second and third bonds, the association rates are equal to $1/\tau$.

### Control experiments
Control experiments were performed by recording 1000 force-distance curves on three different hCD40 transfected CHO cells and three different CHO cells without hCD40 expression at a pulling speed of 3 μm/s. In addition, the hCD40 binding sites for hIgG or hCD40L, coupled to the AFM tip, were blocked with soluble hCD40 by incubating the cantilever tip for 1 hour in HEPES buffer with 1 μM soluble hCD40. Binding probabilities measured on the same hCD40 transfected cells with the same pulling speed were compared before and after the tip block with soluble hCD40. Measurements with hIgG coupled to the AFM tip were performed in HEPES buffer containing 137 mM NaCl, 2.7 mM KCl, 10 mM HEPES (pH 7.4, adjusted with NaOH). Measurements with hCD40L coupled to the AFM tip were performed in the same HEPES buffer with 50 μM NiCl$_2$.

### Competitive binding assay for hIgG and hCD40L
Force-distance curves were recorded on 3 to 4 different hCD40 transfected CHO cells with hIgG1 coupled to the cantilever tip. On each cell, 300 curves were recorded with a pulling speed of 3 μm/

s. Measurements were performed in HEPES buffer containing 137 mM NaCl, 2.7 mM KCl, 10 mM HEPES, and 50 $\mu$M NiCl$_2$ (pH 7.4, adjusted with NaOH). Next, the same measurements were repeated on the exact same cells with hCD40L coupled to the AFM tip. hIgG1 was then added to the buffer of the measurement chamber resulting in a concentration of 0.5 $\mu$M hIgG1. After 30 minutes, 300 force-distance curves were recorded on each of the same 3 to 4 CHO cells with hIgG1 and hCD40L coupled to the AFM tip. Finally, hCD40L was added to the buffer in the measurement chamber resulting in a concentration of 0.5 $\mu$M hCD40L. After 30 min, 300 force-distance curves were recorded on each of the same 3 to 4 CHO cells with hCD40L coupled to the cantilever tip.

### Surface plasmon resonance (SPR)

SPR was performed to determine spontaneous thermodynamic association and dissociation. His-tagged soluble hCD40 was immobilized on Sensor Chip NTA (cytavi) and different concentrations of hIgGs and hCD40L (Acrobiosystems, CDL-H5269) were injected to the measurement chamber of a BiacoreX instrument. All samples were diluted in the HBS running buffer (140 mM NaCl, 5 mM KCl, 10 mM Hepes, pH = 7.4, 0.01% BSA). The recorded binding curves were fitted with the "bivalent model" using Biaevaluation 3.2 RC software (Supplementary Fig. 4, Supplementary Fig. 8, solid line). Furthermore, a "trivalent binding model"[46] was used additionally to characterize the interaction of CD40L and CD40 (Supplementary Fig. 8, dashed line). Kinetic rates and equilibrium constants were determined and are shown in Supplementary Table 1.

### HS-AFM sample preparation and imaging

Stock solutions of soluble hCD40, hCD40L (N-terminal His$_6$-tag) and hIgGs were diluted in HEPES buffer containing 137 mM NaCl, 2.7 mM KCl and 10 mM HEPES (pH 7.4, adjusted with NaOH) to a final concentration of 10 nM. To study hIgG:hCD40 and hCD40L:hCD40 binding dynamics, solutions were prepared by pre-mixing 10 nM of hIgG and 10 nM of hCD40 in a 1:2 ratio and 10 nM of hCD40L and 10 nM of hCD40 in a 1:3 ratio. The sample preparation for all antibodies and proteins studies was the following: 1 $\mu$l of 5 mM NiCl$_2$ solution was applied to a freshly cleaved mica disc (2 mm diameter). After 1 minute of incubation, the mica surface was rinsed 10 times with deionized water. Next, 1 $\mu$l of one of the solutions mentioned above was applied on the mica disc and incubated for 10 minutes. Subsequently, the mica surface was rinsed 10 times with 2 $\mu$l HEPES buffer each. Finally, the sample was inserted into the measurement chamber of the HS-AFM (RIBM, Japan), which was filled with HEPES buffer (137 mM NaCl, 2.7 mM KCl and 10 mM HEPES; pH 7.4). For imaging, ultra-short cantilevers (USC-F1.2-k0.15, Nanoworld) with a nominal spring constant of 0.15 N/m, a resonance frequency in the range of 500 kHz and a quality factor of about 2 were used. The amplitude was set to 85–90% of the free amplitude (about 3 nm) during imaging.

### HS-AFM image processing and volume determination

Gwyddion 2.56 software was used for image processing. The background was subtracted by levelling the image by selection of three points and averaging a small area around the selected points. Next, a Gaussian filter was applied to the image. For volume determination of hCD40, structures above 50% of the height of the raw image were masked and the volume of each individual masked structure was determined. The volume data was represented in an experimental pdf by adding up Gaussians of unitary area entering at the values of their volume (x-axis, Fig. 2b) with widths representing the measurement error of the volume (4 nm$^3$), which was estimated from the volume deviation of the crystal structure. The suchlike constructed pdf therefore consists of the raw volume data (no fit) and can be viewed as the equivalent of a continuous histogram, which has a higher resolution than a conventional histogram and does not suffer from binning artefacts. Subsequently, the pdf was fitted with a multi-Gaussian function and the individual Gaussians representing respective oligomeric states were plotted.

### Reporting summary

Further information on research design is available in the Nature Portfolio Reporting Summary linked to this article.

## Data availability

The crystal structure of the CD40–CD40L complex, as shown in Fig. 2c and Fig. 3f, is available under: 3QD6. Source data are provided with this paper.

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

## Acknowledgements

This work was supported by funding from the Austrian National Foundation for Research, Technology, and Development and Research Department of the State of Upper Austria (Y.J.O.), from the FWF projects V584 (Y.J.O.), P30314 (H.S., P.S., P.H.), P31599 (R.Z.), PIN2697624, P33481-B (B.P.), from the Iris Fischlmayr Research Scholarship of the Johannes Kepler University, Linz, Austria (H.S.) and the ÖAW fellowship STIP13202002 (L.H.). We thank Sabrina Meindlhumer for developing a home-build Matlab program used in this study for SMFS data analysis.

## Author contributions

H.S., P.S. and S.D. performed SMFS measurements and analysed the data. H.S. performed HS-AFM measurements and analysed the data. L.H. performed SPR experiments and analysed the data. F.W. performed fluorescence microscopy measurements and analysed the data. C.W. provided hCD40-transfected CHO cells. R.Z. prepared CHO cells for SMFS measurements. J.CR. provided the different anti-hCD40 hIgG subclasses and soluble hCD40. R.Z., D.B., Y.J.O. and B.P. contributed to study design, data analysis and gave conceptual advice. J.C.R. and P.H. oversaw the conception and supervised the study. H.S. and P.H. wrote the manuscript. All authors read and reviewed the manuscript.

## Competing interests

The authors declare no competing interests.
