## [Transparent Peer Review file · Nature Communications]

Nanomechanical binding mechanism of ligands drives agonistic activity

Corresponding Author: Professor Peter Hinterdorfer

Version 1:

Reviewer comments:

Reviewer #1

(Remarks to the Author)

Comments for the authors

In the manuscript entitled "Nanomechanical binding mechanism of ligands drives agonistic activity", Seferovic et al. describe the use of SMFS and HS-AFM to help understand the binding mechanism of CD40L and a range of agonistic anti-CD40 antibodies to CD40. It is an interesting work, which bring some new mechanistic insight to the field. Unfortunately, much of the analysis is somewhat superficial, with conclusions being made that are not strictly supported by the results. This reduces the impact of the study and dampens the enthusiasm and support for publication in the current form.

The following are major concerns:

1. The use of abbreviations in the manuscript is confusing. I will give a few examples:

Page10 line 179: The abbreviation of "HS-AFM" appears for the first time in the manuscript without any explanation.

The abbreviation SMFS for single-molecule force spectroscopy is sometimes abbreviated (Page 6 Line 124 and 127, Page 7 line 151, etc.) and others spelled in full (Page 4 Line 70 and 72, Page 7 line 147, etc.).

2. Fig. 1C. there are about 5% of trimers, which is colored in red in the pie chart, among the oligomeric state of hCD40. However, the author declares that "Remarkably, hardly trimers or pentamers were found." in line 184 in the article. I don't think the author can make such a conclusion based on the result.

3. The figures of HS-AFM are blurred, especially the parts labeled with antibodies and hCD40L. It is suggested that the authors could fusion different colors of fluorescent proteins on the antibodies and hCD40L, not only CD40, to make the results more clear during performing the relevant experiment.

Reviewer #2

(Remarks to the Author)

Review of the manuscript " Nanomechanical binding mechanism of ligands drives agonistic activity" by Seferovic and coll.

In this study, the authors combined single-molecule force spectroscopy (SMFS) and High-Speed AFM (HS-AFM) on living cells to study the interaction at the single molecule level of trimeric ligand and/or the bivalent antibodies to the CD40. The clustering of CD40 receptors in the membrane of antigen-presenting cells (APC)s, induced by the engagement of CD40 with CD40L, is crucial for their activation and maturation which induces the release of cytokines that stimulate natural killer cells. Recent work has suggested that the activity of antibodies targeting CD40 may vary according to the flexibility of their hinge region, which is inversely correlated with agonistic potency.

This study is part of this context and the authors propose " to paint a detailed dynamic and molecular mechanical picture upon ligation to our antigen candidates, isolated CD40 and CD40-stably expressed in CHO cells". They analyzed the interaction strength and its dynamics during mono or multi-bond formation between CD40L and/or antibodies from SMFS experiments. They attempted to capture the structure of CD40L and/or antibodies, alone or in complex with CD40.

Although I find the results interesting for the field, I have a few major problems listed below. More generally, I find the SMFS data clear and perfectly interpreted but I have more concerns about the HS-AFM data which I find me more questionable.

- The authors identified one or more rupture events during SMFS experiments. The majority of these corresponded to single bonds or double bonds, the latter being assumed to be formed between the two Fab arms of the antibody and two hCD40 molecules. This could be the case but I don't understand how the authors excluded the presence of 2 Fabs per tip which could also explain this behavior (unless they can prove that there is only one Fab per tip). The point should be discussed by the authors.

- I have major problems with the HS-AFM data described in Figure 2 and discussed in the text:

- The way in which the first set of HS-AFM experiments is described and interpreted doesn't really seem appropriate to me. In Figure 2b, different volumes measured by HS-AFM are used to determine the oligomerization state of CD40. However, such a histogram-based classification is very sensitive to the binning used for the volume. It would have been more informative to show the volume measured in a scatter plot. In addition, I don't really understand how these assemblies are organized, for example a tetramer has different shapes in Figure 2a.

- In Figure 2e, HS-AFM images of hlgG1 and hlgG2B are shown in the first column but their shape and volume appear to be altered when interacting with hCD40. The two IgGs should be delineated by a dotted line in all panels.

- On page 10, the text corresponding to Figure 2g indicates that new antigen binding is often accompanied by the movement of one Fab arm, which reinforces the flexibility of the antibody hinge. It looks to me as if the whole Fab molecule has moved during imaging. The authors should draw the lines for the 2 arms as compared to the main axis. The shapes of the antibodies are different in the two panels and the molecule in the second panel is clearly different from what was described in Figure 2e, first column.

- The authors wrote that the classification described above was supported by fluorescence microscopy experiments shown in supplementary movie 4. I don't really agree with this assumption since the resolution of fluorescent microscopy is at least one order of magnitude higher than that of HS-AFM. This movie is not useful.

- The interpretation of HS-AFM images shown in Figure 3f is not clear to me. The authors wrote "Binding of hCD40 to hCD40L occurred on three distinct sites (Figure 3F, second and third row) and was consistent with hCD40L crystal structure depicting the binding epitopes". Although I understand the arrows indicating the binding sites of hCD40L to hCD40, I don't understand the reasoning behind the choice of position. As an example, I have highlighted a part of the imaged structure with a red arrow that could also correspond to hCD40L (see the attached file). This reasoning should be discussed in details to convince the reviewer. In addition, the difference in height obtained for the 2 molecules should be discussed in the context of the model presented in the left panel.

- The model proposed in Figure 5 does not really fit with the data. Indeed, it is written page 21 " This continuous dissociation and re-association mechanism could lead to more and more receptors converging and forming a final cluster (Figure 5 D). I don't think that the data support this idea which could be easy to confirm by adding soluble proteins that should bind to assemblies coated on mica and form clusters as proposed by the authors.

I have also a few minor points

- in the tables, the authors should specify whether the values correspond to SD or EM.

- It should be informative, when plotting the unbinding force as a function of IgG loading (see an example in Figure 1f), to color in red or blue the points corresponding to single or double bonds respectively.

- The title does not really correspond to the results obtained in this study since the SMFS results show that differences in antibody hinge flexibility do not critically affect the formation and lifetime of hlgG:hCD40 single bonds. They do not support the recently published papers described above.

- In Figures 2a, 2c and 2d there are numbers in the panels, but I don't really understand what they mean. While they clearly describe the oligomeric state in Figure 2a, this does not seem to be the case in Figures 2c and 2d.

Version 2:

Reviewer comments:

Reviewer #1

(Remarks to the Author)

Dear authors, thanks for your revision. I don't have extra comments.

Reviewer #2

(Remarks to the Author)

In my opinion, the authors have responded accurately to all my comments and the manuscript is now acceptable for publication in Nature Communications.

Dear reviewer,

we appreciate your responses and have carefully considered the valuable comments and suggestions, making our best efforts to address all concerns in detail and improve the quality of the manuscript. Please see below our point-by-point response to all issues raised in the review. We hope that our manuscript has now reached a publishable format.

Yours sincerely,

Peter Hinterdorfer

Major remarks:

- 1. The use of abbreviations in the manuscript is confusing. I will give a few examples: Page 10 line 179: The abbreviation of “HS-AFM” appears for the first time in the manuscript without any explanation. The abbreviation SMFS for single-molecule force spectroscopy is sometimes abbreviated (Page 6 Line 124 and 127, Page 7 line 151, etc.) and others spelled in full (Page 4 Line 70 and 72, Page 7 line 147, etc.).**

We fully agree with the reviewer that this inconsistency causes confusion for the reader. We have therefore now ensured that abbreviations are defined prior to their first use and that they are used consistently and clearly throughout the manuscript.

- 2. There are about 5% of trimers, which is colored in red in the pie chart, among the oligomeric state of hCD40. However, the author declares that “Remarkably, hardly trimers or pentamers were found.” in line 184 in the article. I don’t think the author can make such a conclusion based on the result.**

We agree and have changed in the manuscript accordingly:

“With 26% of monomers and 23% of dimers hCD40 appeared to be in an equilibrium between the monomeric and dimeric states. Other oligomeric states were less frequent with 8% of trimers, 11% of tetramers and lower proportions for assemblies of higher order (Fig. 2b, pie chart).” – **Page 10, lines 188 - 191**

- 3. The figures of HS-AFM are blurred, especially the parts labeled with antibodies and hCD40L. It is suggested that the authors could fuse different colors of fluorescent proteins on the antibodies and hCD40L, not only CD40, to make the results more clear during performing the relevant experiment.**

Many thanks for pointing this out. To make our results clear, we improved the quality of the images of hCD40, antibodies and hCD40L by increasing the contrast of all images in Fig. 2 and 3 and by converting the images in Fig. 2a and 2e into a 3D representation for better visibility. In addition, all hIlgGs have been marked with a y-shaped outline indicating the characteristic structure of the antibody's Fc region and

its two Fab arms. To indicate monovalent or bivalent binding of hCD40 to hIgG, dashed purple (for monovalent binding) or magenta (for bivalent binding) circles were used to mark the attached hCD40 assemblies (Fig. 2e, f). Similarly, dashed blue circles were used to mark hCD40 bound to hCD40L (Fig. 3f, g). This representation was also applied to the corresponding Supplementary Movies 3-7 of hIgGs and Supplementary Movies 8-12 of hCD40L.

Dear reviewer,

we appreciate your responses and have carefully considered the valuable comments and suggestions, making our best efforts to address all concerns in detail and improve the quality of the manuscript. Please see below our point-by-point response to all issues raised in the review. We hope that our manuscript has now reached a publishable format.

Yours sincerely,

Peter Hinterdorfer

Major remarks:

- 1. The authors identified one or more rupture events during SMFS experiments. The majority of these corresponded to single bonds or double bonds, the latter being assumed to be formed between the two Fab arms of the antibody and two hCD40 molecules. This could be the case, but I don't understand how the authors excluded the presence of 2 Fabs per tip which could also explain this behavior (unless they can prove that there is only one Fab per tip). The point should be discussed by the authors.**

Many thanks for giving us the opportunity to clarify this important point. For our experiments the antibodies were coupled to the AFM cantilever tip via a flexible polyethylene glycol crosslinker at their Fc region. This allows for unconstrained binding of the antibody to the hCD40 receptor and has been shown to facilitate the detection of binding events mediated by the Fab arms of a single antibody coupled to the AFM tip (please see also citation below).

Our experimental results showed unbinding forces in the higher force regime which were in good agreement with theoretical forces for the dissociation of double bonds calculated with the Markov binding model. In addition, the number of two sequential dissociation events was low: $4.08 \pm 0.44\%$ and $2.77 \pm 0.76\%$ (n=2 data sets) for the more flexible hIgG1 and hIgG4, respectively, and lower rates for the less flexible hIgG2 isoforms, $2.42 \pm 2.30\%$ (IgG2B) and $1.48 \pm 1.69\%$ (IgG2A) (n=3 data sets, $p > 0.11$) (page 5, lines 98-100). Both, the absence of higher forces and the low number of sequential dissociation events indicate that only one antibody interacted with hCD40 during our experiments.

Changes in the manuscript:

“This linkage enables unconstrained binding of the antibody paratope to transmembrane hCD40 receptors expressed on the surface of CHO cells in a physiological setting and facilitates the detection of binding events mediated by the Fab arms of a single antibody²⁵.” - **Page 5, lines 83 – 85**

Reference:

Zhu, R. *et al.* Nanomechanical recognition measurements of individual DNA molecules reveal epigenetic methylation patterns. *Nat Nanotechnol* **5**, 788–791 (2010).

- 2. The way in which the first set of HS-AFM experiments is described and interpreted doesn't really seem appropriate to me. In Figure 2b, different volumes measured by HS-AFM are used to determine the oligomerization state of CD40. However, such a histogram-based classification is very sensitive to the binning used for the volume. It would have been more informative to show the volume measured in a scatter plot. In addition, I don't really understand how these assemblies are organized, for example a tetramer has different shapes in Figure 2a.**

As suggested by the reviewer, we changed the representation for determining the number of hCD40 oligomeric states: In detail, volumes of individual molecules were determined as described in the Methods section 4.11. Then, the volume data was represented in an experimental probability density function (pdf, black solid line in Fig. 2b) by adding up Gaussians of unitary area entering at the values of their volume (x-axis) with widths representing the measurement error of the volume (4 nm^3), which was estimated from the volume deviation of the crystal structure. The suchlike constructed pdf therefore consists of the raw volume data (no fit) and can be viewed as the equivalent of a continuous histogram, which has a higher resolution than a conventional histogram and does not suffer from binning artefacts. Subsequently, the pdf was fitted with a multi-Gaussian function and the individual Gaussians representing respective oligomeric states were differently coloured (Fig. 2b). Both, pdf and multi-Gaussian displayed a distinct first (yellow) and second (orange) peak, corresponding to hCD40 monomers and dimers, respectively. Based on the maxima of all peaks (vertical lines in Fig. 2b) volume data points were collected from equidistant segments between two neighbouring peaks and presented in a scatter plot (Fig. 2b), which allowed us to estimate the number of molecules in their oligomeric state: With 26% of monomers and 23% of dimers hCD40 appeared to be in an equilibrium between the monomeric and dimeric states. Other oligomeric states were less frequent with 8% of trimers, 11% of tetramers and lower proportions for assemblies of higher order (Fig. 2b, pie chart).

The oligomeric states of the hCD40 assemblies presented in Fig. 2a were assigned on basis of the calculated volumes. It is known from the literature that hCD40 contains a pre-ligand domain (PLAD) located in cysteine-rich domain 1 (CRD1), which is involved in hCD40 dimerisation^{1,2} and might be the driver for the formation of hCD40 dimers observed in our experiments. However, as higher order hCD40 assemblies were found in various shapes (such as the tetramers in Fig. 2a), we suggest that the formation of higher order assemblies may result from a mechanism other than that initiated by the PLAD.

Changes in the manuscript:

“The volume data was then summed up in an experimental probability density function (pdf), as an equivalent of a continuous histogram representing the original data. The pdf was fitted with a multi-Gaussian function (Fig. 2b, pdf) and the individual Gaussians representing respective oligomeric states were differently coloured (Fig. 2b). Based on the maxima of all peaks (vertical lines in Fig. 2b) volume data points were collected from equidistant segments between two neighbouring peaks and presented in a scatter plot (Fig. 2b), which allowed us to estimate the number of molecules in their oligomeric states: With 26% of monomers and 23% of dimers hCD40 appeared to be in an equilibrium between the monomeric and dimeric states. Other oligomeric states were less frequent with 8% of trimers, 11% of tetramers and lower proportions for assemblies of higher order (Fig. 2b, pie chart).” - **Page 10, lines 182 – 190**

“Experimental probability density function (pdf) and its multi-Gaussian fit of the determined volumes of 111 different hCD40 assemblies showing two pronounced peaks, which can be assigned to monomers (yellow) and dimers (orange), accounting for nearly half of all hCD40 assemblies. The scatter plot depicts all volume data points (grey), which allows for estimating the number of oligomeric states. For each oligomeric state, the mean (colored) and standard deviation is displayed.” – **Figure 2b, page 12, lines 218 - 223**

“The observed dimer formation on mica may be driven by the pre-ligand assembly domain (PLAD) of CD40⁶⁻⁸. However, we found higher order CD40 assemblies in various shapes (Fig. 1a), suggesting that their formation may result from a different mechanism than that initiated by PLAD.” – **Page 10, lines 194 - 197**

“The volume data was represented in an experimental probability density function (pdf) by adding up Gaussians of unitary area entering at the values of their volume (x-axis, Fig. 2b) with widths representing the measurement error of the volume (4 nm³), which was estimated from the volume deviation of the crystal structure. The suchlike constructed pdf therefore consists of the raw volume data (no fit) and can be viewed as the equivalent of a continuous histogram, which has a higher resolution than a conventional histogram and does not suffer from binning artefacts. Subsequently, the pdf was fitted with a multi-Gaussian function and the individual Gaussians representing respective oligomeric states were plotted.” – **Methods 4.11, pages 30-31, lines 604-611**

References:

1. Smulski, C. R. *et al.* Cysteine-rich domain 1 of CD40 mediates receptor self-assembly. *Journal of Biological Chemistry* 288, 10914–10922 (2013).
2. Chan, F. K.-M. *et al.* A domain in TNF receptors that mediates ligand-independent receptor assembly and signalling. *Science* (1979) 288, 2351–2354 (2000).

- 3. In Figure 2e, HS-AFM images of hlgG1 and hlgG2B are shown in the first column but their shape and volume appear to be altered when interacting with hCD40. The two IgGs should be delineated by a dotted line in all panels.**

Fig. 2e, first column, shows the recording of isolated hlgG1 and hlgG2B on a mica surface, whereas in Fig. 2e, second and third column, hlgGs were pre-mixed with hCD40 before incubation on the mica surface. We found that this experimental protocol caused a tip broadening effect, most likely due to soluble hCD40 in the measurement chamber adhering to the cantilever tip. This tip broadening effect resulted in slightly increased dimensions of hlgGs (Fig. 2e, second and third columns) when compared to the recordings of hlgG alone (Fig. 2e, first column).

To improve the visibility of the antibodies, the images in Fig. 2e have been converted to a 3D representation and hlgGs have been marked with a y-shaped outline indicating the characteristic structure of the antibody's Fc region and its two Fab arms. To indicate monovalent or bivalent binding of hCD40 to hlgG, dashed purple (for monovalent bonds) or magenta (for bivalent bonds) circles have been used to mark the attached hCD40 assembly. This presentation has also been applied to Fig. 2f and the corresponding Supplementary Movies 3-7.

Changes in the manuscript:

“Due to a tip broadening effect, possibly caused by soluble hCD40 in the measurement chamber adhering to the cantilever tip, the dimensions of the hlgGs increased slightly (Fig. 2e, second and third columns) in comparison to the recordings of hlgG only (Fig. 2e, first column).” – **Page 10, lines 200 - 202**

“3D representations of HS-AFM images of hlgG2B and hlgG1 without (first column) and with hCD40 (second and third columns). hlgGs formed monovalent (purple) or bivalent (magenta) bonds with different hCD40 oligomers. hlgGs are marked by a y-shaped outline indicating the characteristic structure of the antibody's Fc region and its two Fab fragments.” – **Figure 2e, page 13, lines 226 - 230**

“HS-AFM images of hlgG showing the individual flexibility of the Fab arms (F1, F2) within a certain radius. Purple circles indicate monovalent hlgG:hCD40 bonds, white circles indicate an unbound state. hlgGs are marked by a y-shaped outline indicating the characteristic structure of the antibody's Fc region and its two Fab fragments.” – **Figure 2f, page 13, lines 230 - 234**

- 4. On page 10, the text corresponding to Figure 2g indicates that new antigen binding is often accompanied by the movement of one Fab arm, which reinforces the flexibility of the antibody hinge. It looks to me as if the whole Fab molecule has moved during imaging. The authors should draw the lines for the 2 arms as compared to the main axis. The shapes of the antibodies are different in the two**

panels and the molecule in the second panel is clearly different from what was described in Figure 2e, first column.

As suggested by the reviewer, we have drawn a main axis and introduced a y-shaped outline for the antibody to the image sequence (Fig. 2f) to make it easier to follow the movement of the two Fab fragments (Figure 2f, F1, F2) and the Fc region. The movement of F2 is clearly visible and is accompanied by the association of hCD40 to F2, whereas there is only little movement of F1 or Fc and no displacement of the antibody as a whole. Supplementary Movie 7 also highlights the observed flexibility of F2 as shown in Fig. 2f.

Changes in the manuscript:

“Association and dissociation with surface-immobilized hCD40 molecules, was often accompanied by movement of one Fab arm (Fig. 2g, Supplementary Movie 7).” – **Page 11, lines 211 – 212**

- 5. The authors wrote that the classification described above was supported by fluorescence microscopy experiments shown in supplementary movie 4. I don't really agree with this assumption since the resolution of fluorescent microscopy is at least one order of magnitude higher than that of HS-AFM. This movie is not useful.**

We agree and have therefore removed the results of the fluorescence microscopy experiments from the manuscript.

- 6. The interpretation of HS-AFM images shown in Figure 3f is not clear to me. The authors wrote "Binding of hCD40 to hCD40L occurred on three distinct sites (Figure 3F, second and third row) and was consistent with hCD40L crystal structure depicting the binding epitopes". Although I understand the arrows indicating the binding sites of hCD40L to hCD40, I don't understand the reasoning behind the choice of position. As an example, I have highlighted a part of the imaged structure with a red arrow that could also correspond to hCD40L (see the attached file). This reasoning should be discussed in detail to convince the reviewer. In addition, the difference in height obtained for the 2 molecules should be discussed in the context of the model presented in the left panel.**

Many thanks for giving us the opportunity to depict our representations more clearly. To improve the clarity of Fig. 3f and to emphasise the fact that we only observed either monovalent or bivalent hCD40L:hCD40 binding, but no trivalent hCD40L:hCD40 bond formation, we selected two HS-AFM images in which hCD40L was clearly associated with one hCD40 assembly (Fig. 3f, column 2) and two hCD40 assemblies simultaneously (Fig. 3f, column 3). The association and dissociation of these hCD40 assemblies is clearly visible, as shown in Supplementary Movies 9 and 10. The

selected images indicate that hCD40 could bind to three different binding sites of hCD40L, with only two binding sites occupied simultaneously:

- Binding site 1: Fig. 3f, column 2, Supplementary Movie 9
- Binding site 2: Figure 3f, column 3, Supplementary Movie 10
- Binding site 3: Fig. 3f, column 3, Supplementary Movies 9 and 10

In contrast to the crystal structure (Fig. 3f, first line), hCD40 assemblies were found to be lower in height than the hCD40L to which they were coupled (Fig. 3f, second and third column). This difference in height is attributed to the flexible chain-like structure of hCD40 (Fig. 2c), suggesting that the hCD40 assemblies were not in an upright position, but rather in a lying position on the mica surface during HS-AFM imaging.

Changes in the manuscript:

“Furthermore, in contrast to the crystal structure (Fig. 3f, first line), hCD40 assemblies were found to be lower in height than the hCD40L to which they were coupled (Fig. 3f, second and third column). This difference in height is attributed to the flexible chain-like structure of hCD40 (Fig. 2c), suggesting that hCD40 assemblies were not in an upright position, but rather in a lying position on the mica surface.” – **Page 15, lines 271 - 275**

“Crystal structures (PDB:3QD6) and HS-AFM images of hCD40L only (first column), with one (second column) and two (third column) hCD40 receptors bound. Of the three possible binding sites of hCD40L for hCD40, only a maximum of two were found to be occupied.” – **Figure 3f, Page 17, lines 303 – 306**

- 7. The model proposed in Figure 5 does not really fit with the data. Indeed, it is written page 21 " This continuous dissociation and re-association mechanism could lead to more and more receptors converging and forming a final cluster (Figure 5 D). I don't think that the data support this idea which could be easy to confirm by adding soluble proteins that should bind to assemblies coated on mica and form clusters as proposed by the authors.**

As suggested by the reviewer, we performed further experiments to investigate cluster formation driven by hCD40L or hlgGs and hCD40. By extending the incubation time to 45 min after pre-mixing soluble extracellular hCD40 with hCD40L, we found large hCD40L:hCD40 clusters on the mica surface (Fig. 3h, Supplementary Movie 13), demonstrating that the extracellular domains of hCD40 and hCD40L independently drive cluster formation. In contrast, no such clusters were found for hlgGs using the same experimental protocol.

In addition, our SMFS and SPR measurements showed faster first and second bond formation and faster dissociation of hCD40L compared to the hlgGs studied. It has been noted by others that faster dissociation is a critical factor in driving increased

agonism and may promote faster receptor clustering (pages 21,22, lines 375 - 382): “Counterintuitively, Yu et al.¹ recently found that anti-hCD40 mAbs with lower affinities showed increased agonistic activity and more potent antitumor activity than high-affinity antibodies. The critical factor driving increased agonism was assigned to the higher dissociation rate constants¹. A similar correlation was depicted for anti-Fas antibodies, where the authors proposed⁴ that IgGs first bind receptors bivalent, before one Fab arm dissociates for further recruiting receptors to finally establish a receptor cluster. Antibodies with slow dissociation rates might maintain their bivalent bonds, which impedes them from connecting to other receptors.”

Considering the hCD40L:CD40 cluster formation observed with HS-AFM, our findings on binding kinetics with SMFS and SPR, and findings from the literature, we believe that our data provide good support for the proposed model.

Changes in the manuscript:

“Extending the incubation time to 45 min after pre-mixing soluble extracellular hCD40 with hCD40L, we also found large hCD40L:hCD40 clusters on the mica surface (Fig. 3h, Supplementary Movie 13), demonstrating that the extracellular domains of hCD40 and hCD40L independently drive cluster formation. In contrast, using the same experimental protocol, no such clusters were found for hIgGs.” – **Page 15, lines 282 – 286**

“This idea is supported by our HS-AFM experiments, where we observed large hCD40L:hCD40 clusters, where in contrast no clusters were visible by hIgG.” – **Page 22, lines 404 - 406**

References:

1. Yu, X. *et al.* Reducing affinity as a strategy to boost immunomodulatory antibody agonism. *Nature* 614, 539–547 (2023).
2. Chodorge, M. *et al.* A series of Fas receptor agonist antibodies that demonstrate an inverse correlation between affinity and potency. *Cell Death Differ* 19, 1187–1195 (2012).

Minor remarks:

- 1. In the tables, the authors should specify whether the values correspond to SD or EM.**

We have provided this information for each table in the manuscript and supplementary material.

Changes in the manuscript:

“The errors of k_{off} , k_{on} and k_2 are standard errors derived from least squares fitting. Other errors were calculated using error propagation.” – **Table 1, page 8; Table 2, page 14**

- 2. It should be informative, when plotting the unbinding force as a function of IgG loading (see an example in Figure 1f), to color in red or blue the points corresponding to single or double bonds respectively.**

We considered colouring the corresponding data points for single and double bonds in red and blue, respectively and tested this possibility. However, due to the close proximity and slight overlap of the single and double unbinding forces, we find that colouring the data points makes the representation more confusing.

- 3. The title does not really correspond to the results obtained in this study since the SMFS results show that differences in antibody hinge flexibility do not critically affect the formation and lifetime of hlgG:hCD40 single bonds. They do not support the recently published papers described above.**

We agree and have changed the title accordingly:

“2.1. Association and dissociation rates of hlgG subclasses binding to hCD40” – **Page 5, line 77**

- 4. In Figures 2a, 2c and 2d there are numbers in the panels, but I don't really understand what they mean. While they clearly describe the oligomeric state in Figure 2a, this does not seem to be the case in Figures 2c and 2d.**

Thank you. We have defined the numbers in Figure 2a by adding a figure legend and changed the numbers in Fig. 2c, d and Supplementary Movies 1, 2 to descriptions (monomers, dimers, ...).

f)

g)